# Isolation of Two New Phenolic Glycosides from *Castanopsis chinensis* Hance by Combined Multistep CC and HSCCC Separation and Evaluation of Their Antioxidant Activity

**DOI:** 10.3390/molecules28083331

**Published:** 2023-04-10

**Authors:** Ya-Feng Wang, Ping Lin, Yong-Lin Huang, Rui-Jie He, Bing-Yuan Yang, Zhang-Bin Liu

**Affiliations:** Guangxi Key Laboratory of Plant Functional Phytochemicals and Sustainable Utilization, Guangxi Institute of Botany, Guangxi Zhuang Autonomous Region and Chinese Academy of Sciences, Guilin 541006, China

**Keywords:** phenolic glycosides, *Castanopsis chinensis* Hance, high-speed countercurrent chromatography, antioxidant activity

## Abstract

The characteristics of high polarity and susceptibility to oxidation in phenolic glycosides increase the difficulty of their separation from natural products. In the present study, two new phenolic glycosides with similar structures were isolated from *Castanopsis chinensis* Hance using a combination of multistep CC and high-speed countercurrent chromatography. Preliminary separation of the target fractions was carried out by Sephadex LH-20 chromatography (100–0% EtOH in H_2_O). High-speed countercurrent chromatography with an optimized solvent system of N-Hexane/Ethyl acetate/Methanol/Water (1:6:3:4, *v*/*v*/*v*/*v*) with a satisfactory stationary phase retention and separation factor was used for further separation and purification of the phenolic glycosides. Consequently, two new phenolic glycoside compounds were obtained with purities of 93.0% and 95.7%. 1D-NMR and 2D-NMR spectroscopy, mass spectrometry, and optical rotation were employed to identify their structures, which were assigned as chinensin D and chinensin E. The antioxidant and α-glucosidase inhibitory activities of these two compounds were evaluated using a DPPH antioxidant assay and a α-glucosidase inhibitory assay. Both compounds showed good antioxidant activity with IC_50_ values of 54.5 ± 0.82 µg/mL and 52.5 ± 0.47 µg/mL. The α-glucosidase inhibitory activity of the compounds was poor. The successful isolation and structure identification of the two new compounds provides materials not only for a systematic isolation method of phenolic glycosides with similar structures, but also for the screening of antioxidants and enzyme inhibitors.

## 1. Introduction

The genus *Castanopsis* (D. Don) Spach. comprises more than 120 identified species that are distributed throughout southern and southeastern Asia [1]. There are close to 60 species in China that are distributed throughout the south of the Qinling Mountains [2]. *Castanopsis chinensis* Hance is the dominant species of evergreen broad-leaved forests in the southern subtropical regions of Guangdong, China. *C. chinensis* Hance has economic value due to its hard wood and edible seeds [3,4,5]. Polyphenols, which are the characteristic components of *Castanopsis* plants, have broad cytoprotective effects, such as anti-oxidation, anti-inflammatory, and anti-obesity properties [6,7]. Polyphenols have emerged as potential anti-inflammatory agents or enzyme inhibitors due to their significant biological activity [8,9]. Many plants from this genus *Castanopsis* (D. Don) have been used as Zhuang and Yao medicines for the treatment of swelling and pain, such as trauma and toothache [10,11]. Our previous studies found that *Castanopsis* plants contain a large number of polyphenols, including phenolic glycosides, flavonoids, and triterpenoid tannins, representing novel structures [12,13,14,15,16]. Polyphenols had been demonstrated as bioactive constituents corresponding to the anti-obesity effect of this plant [17,18,19]. In our studies of *Castanopsis* plants, we have found that structurally similar triterpene ellagitannins and phenolic acids are difficult to separate effectively using methods such as chromatographic columns and high-performance liquid chromatography. For example, a large number of triterpene ellagitannins with good anticancer activity were found in the extracts of *Castanopsis fargesii*, but the study of the anticancer mechanism could not be continued because monomeric compounds with high purity and precise structures could not be obtained. A variety of ellagitannins with good tyrosinase inhibiting activity were found in *Castanopsis ceratacantha*, but due to their complicated structure and polarity, only mixtures in the form of isomers could be obtained.

High-speed countercurrent chromatography (HSCCC) is a type of liquid-liquid chromatography with irreversible adsorption, which has a wide range of solvent systems to choose from [20,21], making it widely used for the separation of natural products [22,23]. At present, limited reports have focused on the separation and purification of polyphenols using HSCCC. In this study, HSCCC was applied to separate and purify the phenolic glycosides in *C. chinensis* Hance. The successful isolation of such compounds provided inspiration for the efficient isolation of triterpene ellagitannins and ellagitannins, which were difficult to obtain as monomeric compounds in previous studies.

After preliminary separation by Sephadex LH-20 with a reverse ethanol elution system, compounds of the phenolic glycoside fraction were detected by high performance liquid chromatography, which showed similar retention times under the conditions of the analysis. When the analytical conditions were changed (such as flow rate, proportion of mobile phase, column oven temperature), the compounds still had similar retention times. These results indicated that there were compounds with very similar structures. The complicated compositions and similar structures of the samples were the major difficulties in the separation of phenolic glycosides from *C. chinensis* Hance. In this paper, Sephadex LH-20 with reverse ethanol elution system combined with HSCCC was innovatively used to the achieve rapid separation of phenolic glycosides that could not be effectively separated by HPLC. We also attempted to use crude extracts that were not purified by Sephadex LH-20 with a reverse ethanol elution system for HSCCC separation, but the purities of the obtained compounds were not sufficient for NMR analysis. The experimental results demonstrated that Sephadex LH-20 with a reverse ethanol elution system combined with HSCCC was an effective separation and purification method for polyphenol crude extracts containing multiple impurities. Two new phenolic glycosides were obtained from *C. chinensis* Hance, and a systematic extraction and isolation method of phenolic glycosides from *C. chinensis* Hance was established.

## 2. Results and Discussion

### 2.1. Chromatographic Column Separation

Considering that phenolic glycosides are easily oxidized and soluble in methanol solution, the target components in the fresh leaves of *C. chinensis* (7.5 kg) were extracted by soaking in 80% (30 L) aqueous MeOH at room temperature. As shown in Figure 1, after Sephadex LH-20 CC, MCI gel CHP 20P CC, and Sephadex LH-20 CC (100–0% EtOH in H_2_O) separation, compounds **1**–**2** (17.5 and 17.9 min) were considered as the target compounds. Approximately 2.66 g of mixtures containing compounds **1** and **2** were obtained from 453.4 g of crude extract of *C. chinensis* fresh leaves. Compounds **1**–**2** of the phenolic glycoside fraction were detected by high performance liquid chromatography with Shimadzu Shim-pack GIST C_18_ (250 × 4.6 mm, 5 µm, Shimadzu, Japan), which had similar retention times under the conditions of the analysis. When the analytical conditions were changed (such as flow rate, proportion of mobile phase, column oven temperature), the compounds still had similar retention times. We attempted to obtain both compounds using a semi-preparative liquid phase with Agilent ZORBAX SB C_18_, (250 × 4.6 mm, 5 μm). The purity of the products obtained was not sufficient, as shown in the chromatogram in Figure 2. The structures of the compounds could not be determined from the NMR spectrum.

### 2.2. Selection of Solvent System

It is significant to choose an appropriate biphasic solvent system for the separation of HSCCC. The partition coefficient (K) is a representative value which commonly indicates the ratio of solute distribution in the equilibrated two-phase system. The best K-values of target compounds are usually between 0.5 and 2 [24]. The separation factor α between the two components should be close to 1.5 [24]. Therefore, the characteristics of the solvents including solubility, polarity, viscosity, and specific gravity were comprehensively considered to select an optimal biphasic solvent system for high peak resolution and subsequent separation. According to previous literature, HSCCC also presents advantages in the systematic purification of antioxidants, especially for those with similar structures or polarities. Furthermore, Ming-yuan Cao et al. [25] separated myricitrin, quercitrin, and afzelin from the ethyl acetate fraction of 80% ethanol extracts of *Polygonum multiforum* Thunb by HSCCC. Thus, varieties of biphasic solvent systems consisting of n-hexane-ethyl acetate-methanol-water (Hex-EtOAc-MeOH-H_2_O), each with different proportions, were studied in this work. The K-values and Kα of the target compounds are summarized in Table 1.

### 2.3. HSCCC Separation

Two compounds were successfully separated from the Fr592 by HSCCC, and the corresponding HSCCC chromatograms are shown in Figure 3. The fractions of peaks I and II were collected and dried in a vacuum. The dried samples were re-dissolved with chromatographic methanol and analyzed by HPLC, as described in Section 3.4. The results are shown in Figure 4. The purities of compound **1** (24.5 mg) and compound **2** (57.2 mg) were 93.0% and 95.7%. Their corresponding chemical structures are shown in Figure 5.

### 2.4. Structural Identification

The chemical structures of the two compounds were identified by HR-ESI-MS, ^1^H-NMR, ^13^C-NMR, HSQC and HMBC.

The molecular formula of compound **1** was suggested to be C_30_H_36_O_14_ based on the HR-ESI-MS spectrum with a negative-ion peak at *m*/*z* [M − H]^−^ 619.2084 (calculated 619.2032). Compound **1** had a *β*-D-apiofuranosyl-(1→6)-*β*-D-glucopyranosyl group as the sugar, a 5-*O*-(*E*)-caffeoyl as the ester moiety, and eugenol as the aglycone moiety, based on the similarities of the ^1^H- and ^13^C-NMR spectral data to those of dracunculifosides N [26,27]. The ^1^H-NMR spectrum showed a terminal proton signal at *δ* 5.02 (1H, d, *J* = 7.8 Hz), which showed correlations to two carbon signals at *δ* 77.2 (C-2’) and 146.1 (C-4) in the HMBC experiment (Figure 5). The HMBC experiment displayed long-range correlations between the carbonyl carbon signal of the (*E*)-caffeoyl group (*δ* 168.9) and the H-3 signals of *β*-D-glucopyranose [*δ* 5.17 (1H, t, *J* = 8.9 Hz)] (Figure 5). The above chemical evidence suggested that eugenol was bound to the C-1 position of the *β*-D-apiofuranosyl-(1→6)-*β*-D-glucopyranosyl group, and the (*E*)-caffeoyl group was bound to the C-3 position of the *β*-D-apiofuranosyl-(1→6)-*β*-D-glucopyranosyl group. The structure of compound **1** was therefore established as Chinensin D. The absolute configuration of sugar in compound **1** was identified by HPLC analysis after acid hydrolysis.

Compound **1** was a colorless amorphous powder with the molecular formula of C_30_H_36_O_14_. The HR-ESI-MS showed a [M − H]^−^ ion with *m*/*z* of 619.2084 (calculated 619.2032). The ^1^H-NMR spectrum (500 MHz, Acetone-*d*_6_) displayed signals at *δ* 7.64 (1H, d, *J* = 16.0 Hz, 3’’’), 7.15 (1H, d, *J* = 8.2 Hz, H-5), 7.09 (1H, d, *J* = 1.5 Hz, 5’’’), 6.99 (1H, dd, *J* = 1.5, 8.0 Hz, H-9’’’), 6.85 (1H, d, *J* = 1.5 Hz, H-2), 6.81 (1H, d, *J* = 8.0 Hz, 8’’’), 6.79 (1H, dd, *J* = 1.5, 8.2 Hz, H-6), 6.38 (1H, d, *J* = 16.0 Hz, H-2’’’), 5.99 (1H, m, H-8), 5.17 (1H, t, *J* = 8.9 Hz, H-3’), 5.06 (2H, m, H-9), 5.02 (1H, d, *J* = 7.8 Hz, H-1’), 4.30 (1H, d, *J* = 6.9 Hz, H-1’’), 4.12 (1H, d, *J* = 12.0 Hz, H-6a’), 3.86 (3H, s, H-OCH_3_), 3.83 (1H, m, H-5a’’), 3.81 (1H, m, H-6b’), 3.80 (1H, m, H-4’’), 3.73 (1H, m, H-2’), 3.72 (1H, m, H-4’), 3.71 (1H, m, H-5’), 3.60 (1H, dd, *J* = 6.9, 8.4 Hz, H-2’’), 3.50 (1H, m, H-3’’), 3.45 (1H, d, *J* = 12.2 Hz, H-5b’’), 3.36 (2H, d, *J* = 6.7 Hz, H-7). ^13^C-NMR (125 MHz, Acetone-*d*_6_) *δ*: 136.6 (C-1), 114.1 (C-2), 150.8 (C-3), 146.1 (C-4), 118.5 (C-5), 122.2 (C-6), 40.7 (C-7), 139.1 (C-8), 115.9 (C-9), 56.6 (C-OCH_3_), 102.7 (C-1’), 77.2 (C-2’), 78.4 (C-3’), 73.4 (C-4’), 69.7 (C-5’), 68.9 (C-6’), 104.8 (C-1”), 72.4 (C-2”), 74.1 (C-3”), 69.3 (C-4”), 66.7 (C-5”), 168.9 (C-1”‘), 115.3 (C-2”‘), 147.1 (C-3”‘), 127.7 (C-4”‘), 115.1 (C-5”‘), 149.6 (C-6”‘), 146.8 (C-7”‘), 116.5 (C-8”‘), 122.9 (C-9”‘).

The molecular formula of compound **2** was suggested to be C_31_H_38_O_14_ based on the HR-ESI-MS spectrum with a positive-ion peak at *m*/*z* 657.2154 [M + Na]^+^ (calculated 657.2222). Comparison of the ^1^H-NMR and ^13^C-NMR spectral data of compound **2** with those of compound **1** indicated the presence of eugenol moiety, a 5-*O*-(E)-caffeoyl group, and a *β*-D-glucopyranosyl group in compound **2** [26,27]. The ^13^C-NMR spectra of compound **2** showed one methyl carbon signal (*δ* 17.9) and five aliphatic carbon signals (*δ* 69.8, 72.4, 73.9, 77.6, 102.1) in addition to the signals due to the eugenol, 5-*O*-(E)-caffeoyl, and *β*-D-glucopyranosyl moieties. The five aliphatic carbon signals, one methyl carbon signal, and six proton signals (*δ* 4.71, 3.35, 3.63, 3.69, 3.64, 1.20) indicated the presence of a rhamnose moiety. The HMBC experiment displayed long-range correlations between the carbonyl carbon signal of the (E)-caffeoyl group (*δ* 168.9) and the H-3’ signals of *β*-D-glucopyranose [*δ* 5.14 (1H, t, *J* = 9.0 Hz)], between *δ* 4.91 (1H, d, *J* = 7.8 Hz) of *β*-D-glucopyranosyl moiety and the carbon signals of the eugenol moiety at *δ* 146.1 (C-4), and between the carbonyl carbon signal of the rhamnose moiety *δ* 102.1 (C-1”) and the H-6’ signals of *β*-D-glucopyranose [*δ* 4.0 (1H, d, *J* = 10.6 Hz), 3.63 (1H, m)] (Figure 5). Thus, the above chemical evidence suggested that the eugenol group was bound to the C-1 position of the *β*-D-glucopyranosyl group, the (E)-caffeoyl group was bound to the C-3 position of the *β*-D-glucopyranosyl group, and the rhamnose group was bound to the C-6 position of the *β*-D-glucopyranosyl group. The structure of compound **2** was therefore established and named Chinensin E. The absolute configuration of the sugar was identified by HPLC analysis after acid hydrolysis.

Compound **2** was a colorless amorphous powder with the molecular formula of C_31_H_38_O_14_. The HR-ESI-MS spectrum showed a [M + Na]^+^ ion at *m*/*z* 657.2154 (calculated 657.2222). The ^1^H-NMR spectrum (500 MHz, Methanol-d_4_) displayed signals at δ 6.83 (1H, d, *J* = 1.9 Hz, H-2), 6.08 (1H, d, *J* = 8.2 Hz, H-5), 6.76 (1H, d, *J* = 1.9, 8.2 Hz, H-6), 3.31 (2H, m, H-7), 5.96 (1H, m, H-8), 5.06 (2H, m, H-9), 3.83 (3H, s, H-OCH3), 4.91 (1H, d, *J* = 7.8 Hz, H-1’), 3.82 (1H, m, H-2’), 5.14 (1H, t, *J* = 9.0 Hz, H-3’), 3.77 (1H, m, H-4’), 3.64 (1H, m, H-5’), 4.02 (1H, d, *J* = 9.2 Hz, H-6’), 3.63 (1H, m, H-6’), 6.37 (1H, d, *J* = 16.0 Hz, H-2’’’), 7.62 (1H, d, *J* = 16.0 Hz, H-3’’’), 7.07 (1H, d, *J* = 1.5 Hz, H-6’’’), 6.79 (1H, d, *J* = 8.7 Hz, H-8’’’), 6.97 (1H, d, *J* = 1.5, 8.7 Hz, H-9’’’), 4.71 (1H, s, H-1’’), 3.35 (1H, m, H-2’’), 3.63 (1H, m, H-3’’), 3.69 (1H, m, H-4’’), 3.64 (1H, m, H-5’’), 1.20 (3H, d, *J* = 6.5 Hz, H-6’’). 13C-NMR (125 MHz, Methanol-d_4_) δ: 136.7 (C-1), 114.2 (C-2), 150.8 (C-3), 146.1 (C-4), 118.5 (C-5), 122.1 (C-6), 40.7 (C-7), 138.9 (C-8), 115.9 (C-9), 56.7 (C-OCH3), 102.9 (C-1’), 72.2 (C-2’), 78.5 (C-3’), 73.5 (C-4’), 69.8 (C-5’), 67.4 (C-6’), 102.1 (C-1”), 73.9 (C-2”), 77.6 (C-3”), 72.4 (C-4”), 69.8 (C-5”), 17.9 (C-6”), 168.9 (C-1”‘), 115.3 (C-2”‘), 147.1 (C-3”‘), 127.8 (C-4”‘), 115.1 (C-5”‘), 149.5 (C-6”‘), 146.8 (C-7”‘), 116.5 (C-8”‘), 122.9 (C-9”‘).

### 2.5. Determination of Sugar Configuration

The results of the HPLC chromatogram for the determination of the sugar configuration are presented in Figure 6. The retention times of the three peaks at 18.03–18.32, 29.88 and 28.61 min coincided with derivatives of D-glucose, D-rhamnose, and D-apiose, respectively. The reaction mixture of chinensin D had corresponding signal peaks at retention times of 18.32 min and 28.61 min, and the reaction mixture of chinensin E has corresponding signal peaks at retention times of 18.03 min and 29.88 min. These experimental results showed that the sugars in chinensin D and chinensin E were D-glucose, D-apiose and D-rhamnose, and D-glucose, respectively.

### 2.6. Antioxidant Capacity and α-Glucosidase Inhibitory Activity

The antioxidant activities of chinensin D and chinensin E were investigated using a DPPH assay, with ascorbic acid used as the positive control. The antioxidant capacity of chinensin D and chinensin E are presented in Table 2. The chinensin D and chinensin E showed promising antioxidant activities with IC_50_ values of 54.5 ± 0.82 μg/mL and 52.5 ± 0.47 μg/mL for the DPPH assay. The scavenging activities of chinensin D and chinensin E were weaker than that of ascorbic acid, and the IC_50_ value of chinensin D was equivalent to chinensin E.

Acarbose was used as the positive control in this α-glucosidase inhibitory assay. Among the compounds tested (Table 2), chinensin D and chinensin E showed weaker α-glucosidase inhibitory activities (IC_50_ ≥ 10 mg/mL) compared with the reference acarbose. These findings provide experimental evidence that chinensin D and chinensin E have poor α-glucosidase inhibitory activities.

## 3. Experimental

### 3.1. Reagents and Materials

The *C. chinensis* leaves were collected from Guilin, Guangxi Zhuang Autonomous Region in July 2021, and identified by Professor Yusong Huang (Guangxi Institute of Botany, Guilin, China). A voucher specimen (registration No. 20210703) has been deposited in the Guangxi Key Laboratory of Plant Functional Phytochemicals and Sustainable Utilization Guangxi Institute of Botany, Guilin, China. Column chromatography (CC) was performed using Sephadex LH-20 (25–100 μm; GE Healthcare Bio-Science AB, Uppsala, Sweden) and MCI gel CHP 20P (75–150 μm; Mitsubishi Chemical, Tokyo, Japan). The analytical-grade N-hexane, methanol, ethyl acetate, and anhydrous ethanol were purchased from Xilong Chemical Co., Ltd. (Guangdong, China). The HPLC-grade ACN was provided by Spectrum Chemical (Shanghai, China). Acetone-*d_6_* and Methanol-*d_4_* were used as solvents for NMR analysis. Deionized water purchased from Wahaha Group Co., Ltd. was used throughout the experiment. Amberlite IRA400 (Shanghai Macklin Biochemical Co., Ltd., Shanghai, China) was also used, along with DPPH (Macklin; Shanghai, China), ascorbic acid (Sigma, Darmstadt, Germany), *p*-nitrophenyl-*α*-D-glucopyranoside (Sigma), *α*-glucosidase(Sigma), and Acarbose (Sigma). Analytical-grade pyridine, L-cysteine methyl ester hydrochloride, and o-torylisothio-cyanate were purchased from Shanghai Macklin Biochemical Co., Ltd. D-glucose standard (Macklin; Shanghai, China), D-apiose standard (Macklin; Shanghai, China), D-rhamnose standard (Macklin; Shanghai, China) were also used.

### 3.2. Extraction and Isolation of Phenolic Glycosides

The fresh leaves of *C. chinensis* (7.5 kg) were extracted with 80% (30 L) aqueous MeOH (three times, 4 days for each time) at room temperature. The combined solvent was evaporated under reduced pressure to yield a crude extract, which was suspended in water and then extracted with petroleum ether three times. The resulting aqueous solution was subjected to Sephadex LH-20 CC (10 cm × 100 cm, 0–100% MeOH in H_2_O) to yield nine fractions. Fraction 5 (71.2 g) was separated by MCI gel CHP 20P CC (8 cm × 100 cm, 0–100% MeOH in H_2_O) to give nine fractions. Fr5-9 (6.1 g) was further separated by Sephadex LH-20 CC (4.5 cm × 50 cm, 100–0% EtOH in H_2_O) to yield two fractions. Fraction Fr592 (2.7 g), which displayed as a single dot in ferric chloride ethanol developer (1% ferric chloride ethanol solution), was then subjected to further HSCCC separation.

### 3.3. HSCCC Separation

#### 3.3.1. Selection of Two-Phase Solvent System

A 5 mL solvent system was configured according to the ratio list in Table 1, and 10 mg of crude extract (Fr592) was added. The lower phase was directly analyzed by HPLC, and the upper phase was evaporated under reduced pressure to yield a crude extract, which was then reconstituted with an equal volume of methanol for HPLC analysis. The partition coefficients (K-values) were calculated as the peak area of the target compound in the upper phase divided by the peak area of the target compound in the lower phase. Consecutive partition coefficient (Kα) values were calculated for each pair of separated compounds 1 and 2 in each solvent system tested.

#### 3.3.2. HSCCC Separation Procedure

Approximately 2000 mL of the solvent system of N-Hexane/Ethyl acetate/Methanol/Water (1:6:3:4, *v*/*v*/*v*/*v*) was prepared. The mixed solvent was degassed using an ultrasonic bath for 20 min. For HSCCC separation, the column was first filled entirely with the upper phase (stationary phase), and the lower phase (mobile phase) was then pumped into the column at a flow rate of 2.0 mL/min (head to tail). The apparatus was rotated at 860 rpm in reverse mode. The separation temperature was 25 °C. The detection wavelength was 227 nm. Approximately 100 mg of the crude extract powder was dissolved in 10 mL of the solvent system and then injected into the column. After separation, two fractions named fractions I and II were collected. The fractions were condensed under a vacuum at 50 °C and reconstituted with chromatographic methanol for HPLC analysis as described in Section 3.4.

### 3.4. HPLC Analysis

The HPLC experiment was conducted using a Shimadzu LC-20AT (Tokyo, Japan) and a Shimadzu Shim-pack GIST C_18_ (250 × 4.6 mm, 5 µm, Shimadzu, Japan) at a flow rate of 1.0 mL/min. The detection wavelength was 227 nm and the column temperature was 30 °C. The mobile phase was a mixture of ACN (A)-water (B) with a gradient elution program as follows: 0–30 min, 5−95% A. The sample injection volume was 10 μL.

### 3.5. Identification of Target Compounds

The target compounds were dissolved by acetone-*d*_6_ or methanol-*d*_4_ and identified by ^1^H, ^13^C-NMR, ^1^H-^1^H COSY, HSQC, and HMBC spectroscopy. The molecular weights were obtained by high-resolution mass spectrometry equipped with an ESI resource. The negative mode was selected, and the spray voltage was 2800 V. The sheath gas pressure was 40 psi and the aux gas pressure was 10 psi. The capillary temperature was 320 °C. The scanning range of *m*/*z* was from 100 to 1000.

### 3.6. Spectroscopic Data

Chinensin D is a colorless amorphous powder with a molecular formula of C_30_H_36_O_14_. Its specific optical rotation is [α]^25^_D_ = −10.8^◦^(c = 0.1, MeOH). Its 1D and 2D NMR spectra are available in the Appendix A. The HR-ESI-MS result showed a *m*/*z* of 619.2084 [M − H]^−^ (calculated, 619.2032) (Appendix A).

Chinensin E is a colorless amorphous powder with a molecular formula of C_31_H_38_O_14_. Its specific optical rotation is [α]^25^_D_ = −33.1^◦^(c = 0.1, MeOH). Its 1D and 2D NMR spectra are available in the Appendix A. The HR-ESI-MS result showed a *m*/*z* of 657.2154 [M + Na]^+^ (calculated, 657.2222) (Appendix A).

### 3.7. Determination of Sugar Configuration

The sugar configuration was determined following the protocol outlined in the literature [28]. The compound (1.0 mg) was hydrolyzed by heating in 0.5 M HCl (0.6 mL) at 90 °C for 2 h. The residue was then fully dryied in vacuo after neutralizing with Amberlite IRA400. The residue was dissolved in pyridine (0.2 mL) containing L-cysteine methyl ester hydrochloride (1.0 mg) and heated at 60 °C for 1 h. Pyridine (0.2 mL) containing o-torylisothio-cyanate (1.0 mg) was added to the mixture, then heated at 60 °C for 1 h. The standards D-glucose, D-rhamnose, and D-apiose were derivatized using the same method, and a reaction mixture without sugar was used as a blank control. All reaction mixtures were directly analyzed by reversed-phase HPLC. The retention time of the three peaks at 18.03–18.32, 29.88, and 28.61 min were consistent with derivatives of D-glucose, D-rhamnose, and D-apiose.

### 3.8. Antioxidant Activity

The antioxidant ability of the new compounds was performed by the DPPH• assay, as previously described [29]. Briefly, pure compounds (1.0 mg/mL) and different concentrations (3.91, 7.82, 15.63, 31.25, 62.50 μg/mL) of the standard compound (ascorbic acid) were prepared. The sample solution was diluted to a series of appropriate concentrations. Next, 0.10 mL of diluted sample/standard solution was added to 0.30 mL of 0.21 mM DPPH solution and incubated for 30 min in the dark at room temperature. After incubation, the absorbance was measured at 517 nm using a Shimadzu UV-2401A spectrophotometer. Ascorbic acid was used as the positive control. A blank control was prepared by replacing the sample solution with an equal amount of 95% ethanol. The DPPH radical inhibition ability was determined using the following formula. Half maximal inhibitory concentration (IC_50_) values were calculated based on three independent experiments.
inhibition = [Ablank control − Asample]/Ablank control
Ablank control = absorbance value of DPPH blank (DPPH + ethanol) solution
Asample = absorbance of sample solution (sample+ ethanol) solution

### 3.9. α-Glucosidase Inhibitory Assay

The *α*-glucosidase inhibitory activity of the new compounds was evaluated using a previously described procedure [30,31]. First, 50 μL of 0.01 M phosphate buffer (pH = 7.0) was added into a 96-well plate, followed by adding 10 μL of *α*-glucosidase (1 U/mL) in 0.01 M phosphate buffer (pH = 7.0) to the sample solution (20 μL), and then the mixture was incubated for 5 min at 37 °C. Then, 20 μL of *p*-nitrophenyl-*α*-D-glucopyranoside (PNPG, 1 mM) was added to start the reaction. Each reaction was carried out at 37 °C for 30 min and stopped by adding 50 μL of Na_2_CO_3_ (0.2 M). After the reaction, the absorbance was measured at 405 nm. The blank control was prepared by replacing the sample solution with an equal amount of phosphate buffer. Enzymatic activity was quantified using the following formula. Half maximal inhibitory concentration (IC_50_) values were calculated based on three independent experiments.
inhibition = [Asample − Asample control]∕Ablank control
Asample = absorbance of sample reaction (sample+ α-glucosidase+ PNPG) solution
Asample control = absorbance value of sample control (sample+PNPG) solution
A_blank control_ = absorbance value of blank control (*α*-glucosidase+ PNPG) solution

## 4. Conclusions

Polyphenols are known as a main component of *C. chinensis*, but they had not been systematically studied. In the present paper, two new phenolic glycosides with similar structures were separated from *C. chinensis* by combining multistep column chromatography and HSCCC. Their structures were identified as chinensin D and chinensin E by 1D-NMR, 2D-NMR, HR-ESI-MS, and OR. Chinensin D (24.5 mg) and chinensin E (57.2 mg) were obtained from the Fr592 fraction (100 mg) by one-step HSCCC separation with purities of 93.0 and 95.7%, respectively. This paper provided practical experience for the separation of phenolic glycosides with similar structures by combining multistep column chromatography and HSCCC. This experiment proved that Sephadex LH-20 CC (100–0% EtOH in H_2_O) with reverse elution could effectively separate compounds with different molecular weights from similar polarity compounds, which was helpful for the separation of polyphenolic compounds. HSCCC chromatography could be considered if a single color-emitting spot was detected by TLC, but the NMR spectrum showed that the compound was not pure enough. The separation methods used in this paper were highly instructive for the difficult problem of obtaining monomeric compounds of high purity in natural product research. The results of the antioxidant assays showed that chinensin D and chinensin E had strong antioxidant activity. These two compounds could potentially be used as antioxidants and provide a basis for further studies on *C. chinensis* leaves.

## Figures and Tables

**Figure 1 molecules-28-03331-f001:**
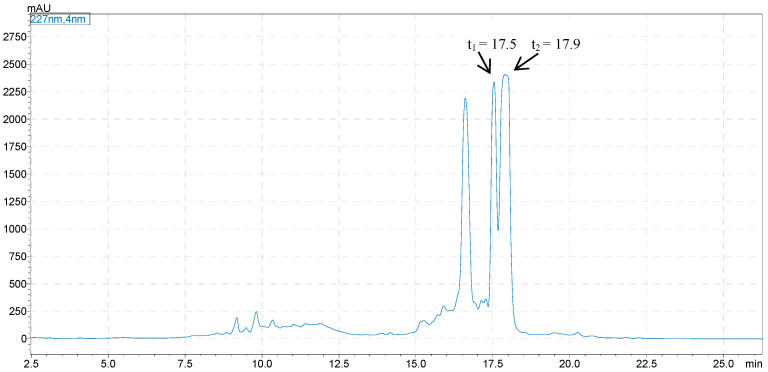
HPLC chromatogram of the crude extract of *C. chinensis* after enrichment by multistep column chromatography (MCC).

**Figure 2 molecules-28-03331-f002:**
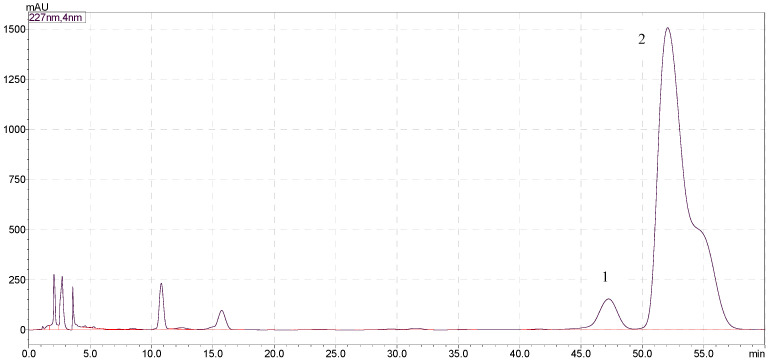
Chromatogram of the semi-preparative liquid phase with Aiglent ZORBAX SB C_18_, (250 × 4.6 mm, 5 μm), mobile phase: ACN (A), water (B), 0–60 min, 22% A; flow rate: 1.0 mL/min; detection: 227 nm.

**Figure 3 molecules-28-03331-f003:**
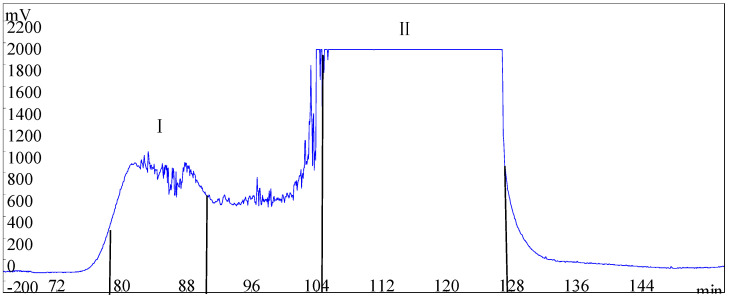
High-speed countercurrent chromatography (HSCCC) chromatogram of the crude extract of Fr592 enriched by multistep column chromatography. Solvent system: N-Hexane/Ethyl acetate/Methanol/Water (1:6:3:4, *v*/*v*/*v*/*v*); mobile phase: the upper phase; flow rate: 2.0 mL/min; revolution speed: 860 rpm; detection wavelength: 227 nm; sample size: 100 mg; injection volume: 10 mL; retention of stationary phase: 53.2%.

**Figure 4 molecules-28-03331-f004:**
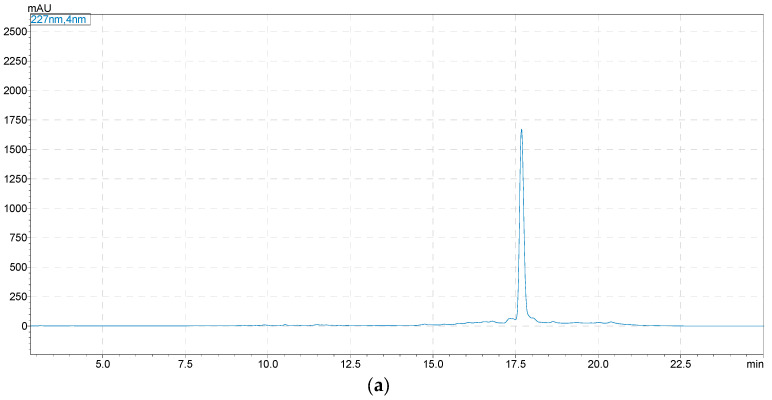
HPLC chromatogram of high-speed countercurrent chromatography (HSCCC) fractions for compound **1** (**a**) and compound **2** (**b**). Analytical column: Shimadzu Shim-pack GIST C_18_ (250 × 4.6 mm, 5 µm, Shimadzu, Japan); mobile phase: ACN (A)-water (B), 0–30 min, 5–95% A; flow rate: 1.0 mL/min; detection: 227 nm.

**Figure 5 molecules-28-03331-f005:**
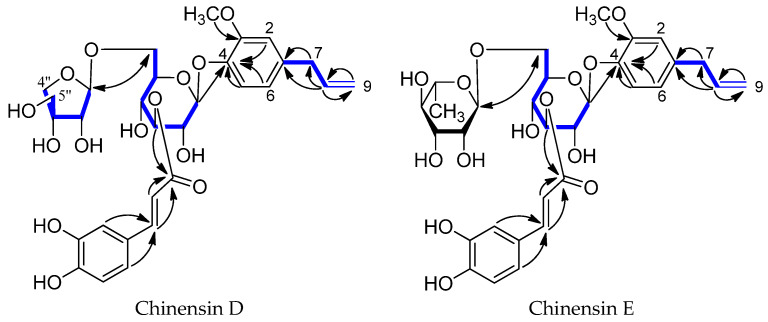
Structure and selected HMBC, ^1^H-^1^H COSY correlations of Chinensin D and Chinensin E.

**Figure 6 molecules-28-03331-f006:**
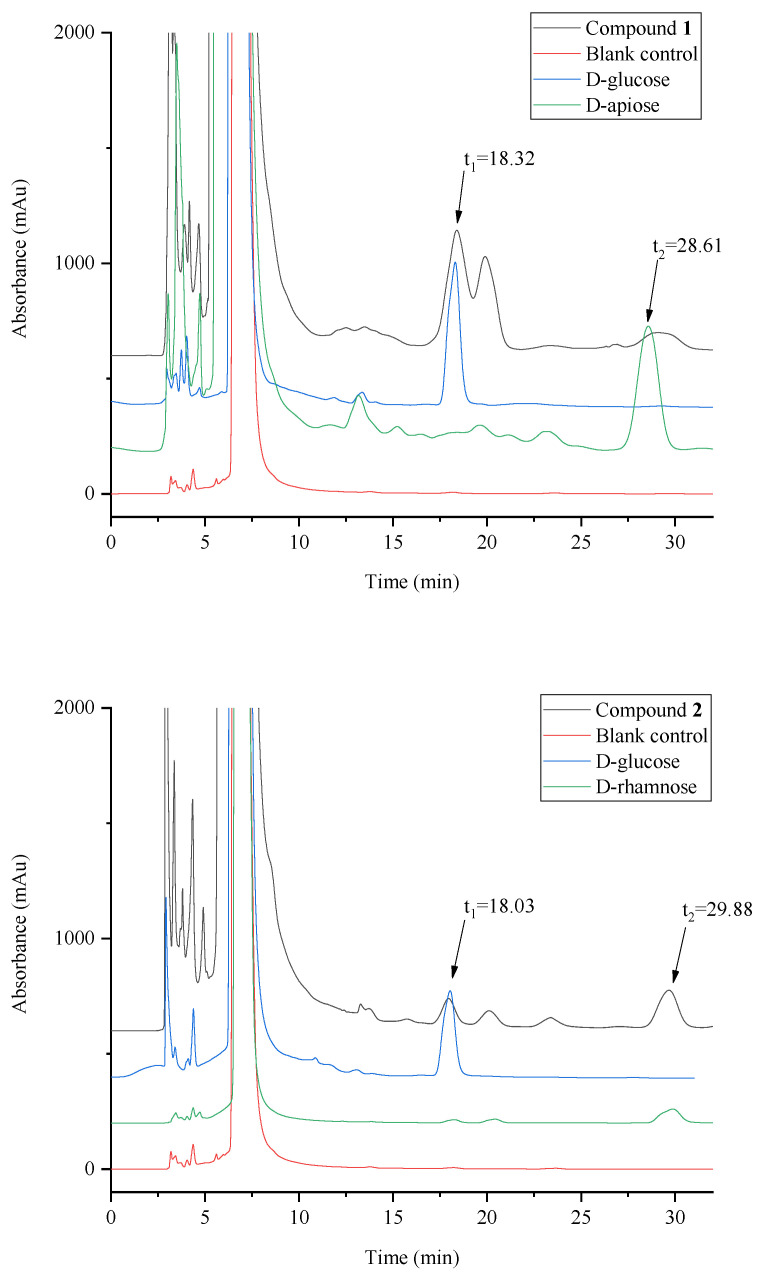
HPLC chromatogram for the determination of sugar configuration. Analytical column: Shimadzu Shim-pack GIST C_18_ (250 × 4.6 mm, 5 µm, Shimadzu, Japan); mobile phase: 25% CH_3_CN in water; flow rate: 0.8 mL/min; detection: 254 nm.

**Table 1 molecules-28-03331-t001:** K and Kα values of target compounds **1** and **2** measured in the solvent systems tested.

No.	Solvent Systems (*v*/*v*/*v*/*v*)	K1	K2	Kα 1/2
1	N-Hexane/Ethyl acetate/Methanol/Water 1:6:3:3	0.34	0.69	2.03
2	N-Hexane/Ethyl acetate/Methanol/Water 1:6:3:4	0.51	0.83	1.63
3	N-Hexane/Ethyl acetate/Methanol/Water 1:6:4:3	0.10	0.09	1.11

**Table 2 molecules-28-03331-t002:** Antioxidant and α-glucosidase inhibitory activities of chinensin D and chinensin E. Mean value ± standard deviation (*n* = 3).

Compounds	DPPH IC_50_ (µg/mL)	α-glucosidase IC_50_ (mg/mL)
chinensin D	54.5 ± 0.82	≥10
chinensin E	52.5 ± 0.47	≥10
Ascorbic acid	13.1 ± 0.21	-
Acarbose	-	0.515 ± 0.08

## Data Availability

The data that support the findings of this study are available from the corresponding author upon reasonable request.

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
