# Peer review of "Isolation of Two New Phenolic Glycosides from Castanopsis chinensis Hance by Combined Multistep CC and HSCCC Separation and Evaluation of Their Antioxidant Activity"

_molecules, 2023, doi:10.3390/molecules28083331_

Round 1

Reviewer 1 Report

The presented problem is interesting but there are significant remarks

1. Novelty of the work is not presented well.

2. Section 2.1. Chromatographic column separation is not presented enough. Add results of all investigated columns.

3. Table 1 does not present several solvent systems. It presents only ratio of one sovent system

4. Describe choice of chromatographic conditions with obtained results.

5. In Fig 3 - 2 chromatograms. Describe both

6. Conclusion should be rewritten.

Author Response

The responses to the reviewers are in the document below

Reviewer 2 Report

The topic of the Manuscript is interesting however some parts are not satisfactory written and must be improved as suggested below;

Fig. 1 – Caption; should be; „HPLC chromatogram of the extract of C. chinensis enriched by multistep column chromatography (MCC). It is my suggestion to introduce this abbreviation of the multistep CC separation before HSCCC.

As MCC we understand several steps of Sephadex LH-20 CC, followed by MCI gel CHP 20P CC and next Sephadex LH-20 CC with reverse elution.

This should be mentioned in the text that not only one CC separation was done before HSCCC.

In the Line 256 – it was suggested that after HSCCC semipreparative HPLC was done. However, in the text we have no information about semipreparative HPLC conditions. – please clarify this inconsistency

In this context the title not adequately describes what was done during experiments, as HSCCC separation was one of the steps. The title as follows better suits to this article; “Isolation of two new phenolic glycosides from Castanopsis chinensis Hance by combined multistep CC and HSCCC separation and evaluation of their antioxidant activity”

On the Figure 1 it is not clear where are the compounds 1 and 2 (tR = 17.5 and 17.6 ?) – please indicate them clearly, because numbers 1 and 2 are above very small peaks tR c.a. 12,8 min.

Section 3.2; It was mentioned that Fr592 was subjected for HSCCC separation – it is not clear why “2” was added at the end of the number. We know as follows; Fr 1-9 from first CC separation, Fr 5 from this was separated by MCI gel CHP 20P CC to give fractions 5/1 – 5/9, and finally Fr 5/9 was separated on reverse Sephadex CC to give how many fractions ? As we can deduced Fr 5/9/2 was separated by HSCCC – If yes why do not describe this clearly ?

Section 2.1. ; lines 67-69 – yield in % should be given (yield in mg/g ; 5.9 mg/g means 0.59 g/100 g = 0.59 % and it is not very high). The description of the amounts of extracts is not clearly presented – it should be rewritten to be more clear. As I found in the text it was as follows; 453.4 g of the crude methanol (80%) extract was collected from 7.5 kg of leaves (fresh weight) . After MCC separation it was finally 2.66 g of the fraction FR5/9/2 which could be subjected for HSCCC (?) In one step 100 mg of this fraction was separated by HSCCC (?) Am I right ?

What was the amount of the compound 1 and compound 2 collected in the single run of HSCCC ?

Line 53-54; "compounds show the small split" - should be "compounds have similar retention times under the conditions of the analysis". It is possible that under other conditions, e.g. in different mobile phases, these compounds could be separated to the baseline and quantified in the extract - has this been tested?

What was the amount of target compounds in the leaves of the C. chinensis plant on a dry weight basis in mg? Has it been analyzed?

Line 231; Was the aqueous solution concentrated before CC Sephadex separation?

Line 219; How is deionized water obtained? Was any deionized water system used?

Section 3.3.1. and Table 1;

Since the authors calculated the K values (K - partition coefficient values) for each compound between the two phases in the 3 solvent systems, this is shown in Table 1. However, also the partition coefficients Kα between compounds 1 and 2 should be calculated and included in Table 1 as an additional column.

These values (calculated from the data in Table 1) are approximately as follows; for solvent system 1 = 2.0; solvent system 2 = 1.6; solvent system 3 = 1.1.

Since a satisfactory value of Kα should be around 1.5 for separation of target compounds, system 2 is fine. Please calculate the Kα values to 2 decimal places and place them in Table 1. The title of the column should be "Kα 1/2". The title of the table should be "K and Kα values of target compounds 1 and 2 measured in the solvent systems tested". As you can see, only three solvent systems were tested (?) Why only these systems - were any literature data inspired? This could be an interesting part of the discussion.

In the content of point 3.3.1. this should be stated as follows; "Consecutive partition coefficient (Kα) values were calculated for a pair of separated compounds 1 and 2 in each solvent system tested."

Minor issues;

Some parts of the text need to be rewritten to keep the same style; see lines 278-282 and lines 291-293 and lines 297-298,

Some passages are not clear and require rewriting to be more concise; e.g. lines 35-37 and Concluding Remarks section.

Line 307; it should be; “…to the previously described procedure [22,23].”

Line; 312; it should be; “After the reaction the absorbance was measured at 405 nm.”

Line 322-323; it should be “…by combining multistep column chromatography and HSCCC…”

Line 53; it should be “…high performance liquid chromatography…”not “high phase liquid chromatography …”

Lines ; 22, 98, 99, 104, 190, is “ml” – it should be “mL”

Line 180 it is “fig.5” should be “Fig.5.”

Line 41 it should be “…representing novel structures…”

Round 2

Reviewer 1 Report

Not all comments were taken for consideration.

1. Novelty of the work is not presented well.

2. Section 2.1. Chromatographic column separation is not presented enough. Add results of all investigated columns. I mean HPLC columns

3. In Fig 3,5 add exact column'

4. Read and analyse all mauscript carefully. Originality / Novelty and Quality of Presentation are low.

Round 3

Reviewer 1 Report

Authors modified manuscript but still have some issue for correction

1. In my opinion, novelty of work is not enough presented.

2. Authors should add more references and describe them

Author Response

Comments and Suggestions for Authors

In my opinion, novelty of work is not enough presented.
Authors should add more references and describe them

Different comparative experimental results have been added, all of which show that the separation method used in the article is effective. This method has reference value for similar problems in natural products. 
Red marked in the article

More references have been added regarding method selection and preliminary experiments.